# Cell-Type-Specific Mitochondrial Quality Control in the Brain: A Plausible Mechanism of Neurodegeneration

**DOI:** 10.3390/ijms241914421

**Published:** 2023-09-22

**Authors:** Hariprasath Ragupathy, Manasvi Vukku, Sandeep Kumar Barodia

**Affiliations:** Centre for Brain Research, Indian Institute of Science, Bengaluru 560012, India

**Keywords:** astrocytes, microglia, mitochondria, neurons, oligodendrocytes, oxidative stress

## Abstract

Neurodegeneration is an age-dependent progressive phenomenon with no defined cause. Aging is the main risk factor for neurodegenerative diseases. During aging, activated microglia undergo phenotypic alterations that can lead to neuroinflammation, which is a well-accepted event in the pathogenesis of neurodegenerative diseases. Several common mechanisms are shared by genetically or pathologically distinct neurodegenerative diseases, such as excitotoxicity, mitochondrial deficits and oxidative stress, protein misfolding and translational dysfunction, autophagy and microglia activation. Progressive loss of the neuronal population due to increased oxidative stress leads to neurodegenerative diseases, mostly due to the accumulation of dysfunctional mitochondria. Mitochondrial dysfunction and excessive neuroinflammatory responses are both sufficient to induce pathology in age-dependent neurodegeneration. Therefore, mitochondrial quality control is a key determinant for the health and survival of neuronal cells in the brain. Research has been primarily focused to demonstrate the significance of neuronal mitochondrial health, despite the important contributions of non-neuronal cells that constitute a significant portion of the brain volume. Moreover, mitochondrial morphology and function are distinctly diverse in different tissues; however, little is known about their molecular diversity among cell types. Mitochondrial dynamics and quality in different cell types markedly decide the fate of overall brain health; therefore, it is not justifiable to overlook non-neuronal cells and their significant and active contribution in facilitating overall neuronal health. In this review article, we aim to discuss the mitochondrial quality control of different cell types in the brain and how important and remarkable the diversity and highly synchronized connecting property of non-neuronal cells are in keeping the neurons healthy to control neurodegeneration.

## 1. Introduction

Mitochondrial dysfunction and oxidative stress are considered as the key central mechanisms of progressive cell loss in several neurodegenerative diseases. Neurons are solely dependent on oxidative phosphorylation to maintain their energy requirements during healthy conditions. In the nervous system, mitochondria regulate neurite branching and regeneration [1], as well as synaptic strength, stability and signaling [2]. Moreover, mitochondrial stress related to mtDNA (mitochondrial DNA) dysregulation can produce neuronal dysfunction and death via impaired ETC (electron transport chain) activity, which results in deficient ATP (adenosine triphosphate) production and related increases in mitochondrial ROS (reactive oxygen species) production. Therefore, mitochondrial quality control and bioenergetic demand in the CNS (central nervous system) are important factors to maintain a healthy brain. The mitochondrial anchor protein SNPH (syntaphilin) is a key mitochondrial protein normally expressed in axons to maintain neuronal health by positioning mitochondria along axons for metabolic needs [3]. In neurons, cofilin1, a key regulator of actin dynamics, may contribute to degenerative processes through the formation of cofilin-actin rods, and through enhanced mitochondrial fission, mitochondrial membrane permeabilization, and the release of cytochrome C. Overall, mitochondrial impairment induced by the dysfunction of actin-regulating proteins such as cofilin1 emerge as important mechanisms of neuronal death with relevance to acute brain injury and neurodegenerative diseases [4]. Although neurons are predominantly highly specialized cells in the brain due to their unique property of transmitting, processing, and storing information, they are entirely dependent on supportive glial cells for their bioenergetic demand (Figure 1). Non-neuronal cells support essential brain functions, such as maintenance of neurotransmitter communication, metabolism, trophic support, formation of myelin sheath, wound healing and immune surveillance [5].

Nonetheless, mitochondrial dysfunction in non-neuronal cells (astocytes, microglia and oligodenrocytes) and its influence on neuronal activity and brain function are poorly studied. Astrocytes and microglia remain functionally healthy even after the loss of their mitochondrial function, due to their higher antioxidant capacity than neurons, primarily on account of glycolysis to produce energy. Astrocytes are the major secretory cells characterized by their defined characteristic of releasing gliotransmitters, including neurotransmitters. Astrocytic factors assist neurons in keeping normal physiological functioning and maintenance of the cerebral vasculature. Also, astrocytes play an important role in synaptic pruning due to their phagocytic property [6,7]. To prevent any inflammatory impact on neurons, microglia act as the first line of defense. In addition, microglial activation is a dynamic event governed by the healthy mitochondria population playing a central role in neuroinflammation, which can be beneficial or pathogenic for neuronal health. In response to a pathogen, microglia act quickly by acquiring different phenotypes to defend neurons. During neuroinflammation, reactive microglia and astrocytes secrete soluble mediators that adversely impact neuronal health and promote neurodegeneration. Mitochondrial dysfunction has been proposed as a key element of α-syn (alpha-synuclein)-induced toxicity in neurons as well as in glial cells. Most of the studies are primarily focused towards understanding α-syn-induced neurotoxicity and its role in neurodegenerative diseases. However, recent evidence showed that non-neuronal cells are involved in the initial phase of pathogenesis. A close association between glial mitochondria and synucleinopathy could be a novel mechanism implicating the roles of glial cells in α-syn-mediated neurodegeneration [8]. Oligodendrocytes are myelinating cells of the brain, keeping the neurons healthy. Mitochondrial dysfunction has been suggested as the main cause of oligodendrocyte damage during neuroinflammation, as observed in MS (multiple sclerosis). NCX (Na^+^-Ca^2+^ exchanger) constitutes the main cellular Ca^2+^ flux system in neurons and microglia. Immunohistochemical studies demonstrated that in A53T mutant mice, NCX1 was abundantly expressed in microglia in the striatum, whereas NCX3 expression was reduced in dopaminergic neurons in SNpc. In vivo and cultured cell studies suggest that the downregulation of NCX3 protein levels and its diminished activity may induce dysfunctional mitochondria and neurotoxicity, whereas NCX1 upregulation in microglia may enhance their production in the striatum [9]. Even though few cell-type-specific roles of mitochondria are accomplished by altering the metabolic functions of mitochondria, unique mitochondrial features were reported in immune or redox signaling [10], and likely many more remain elusive. Accordingly, mitochondria in different cells or cell types, or even in different parts of the same cell, can behave quite differently.

## 2. Mitochondrial Quality Control in Neurons: A Promising Mechanism of Neurodegeneration

### 2.1. Mitochondrial Dynamics in Neurons

Neuronal health is primarily governed by the normal structure and functioning of mitochondria. A systematic network of cellular surveillance mechanisms defends mitochondria against stress by specifically targeting the mitochondrial damage and confirming the selective elimination of dysfunctional mitochondrial proteins or organelles through a highly specialized intracellular event termed “mitophagy”. Moreover, mitochondrial dysfunction is a key player in glutamate-evoked neuronal excitotoxicity. The fission/fusion dynamics of mitochondria are crucial for mitochondrial structure and function. Actin dynamics regulate mitochondrial morphology and function. Inducible excision of the Drp1 protein triggers stress signaling in neurons, which, in turn, activates a cumulative stress response to diminish the expression of the cytokine Fgf21 [11]. Another study supporting the inevitable role of Drp1 demonstrated that Drp1 ablation in neurons of mouse forebrain leads to altered mitochondrial morphology in the hippocampus, which is less effective to antioxidant exposure [12]. Under stress conditions, another key mitochondrial mechanism, i.e., the Mul1-Mfn2 pathway, regulates neuronal mitochondrial quality, where Mul1 deficiency elevates Mfn2 that induces mitochondrial hyperfusion acting as an ER-Mito tethering inhibitor. This overall reduction in ER-Mito connection triggers an increase in cytosolic Ca^2+^ content that, in turn, stimulates calcineurin and causes Drp1-dependent mitochondrial fission and mitophagy. This tightly regulated mitochondrial morphology and ER-Mito interactions play an early key checkpoint role in maintaining mitochondrial quality [13]. Defective mitochondrial biogenesis is another causal event that leads to elevated oxidative stress and neuronal death. Mfn2 removal and mitochondrial fission in mature neurons induce neurodegeneration due to elevated oxidative stress and neuroinflammation [14]. In motor neurons, glutamate excitotoxicity induces the accumulation of aberrant mitochondria by disrupting mitochondrial turnover via Mfn2 degradation by calpain [15]. Such defects result from the PARIS (a novel KRAB and zinc finger protein, substrate of Parkin) -dependent repression of dopaminergic PGC-1α and transcription factors NRF1 and TFAM, which, together, promote mitochondrial biogenesis [16].

### 2.2. Mitochondrial Bioenergetics

Similarly, neuronal health depends on normal ATP production via oxidative phosphorylation. Under normal physiological conditions, there exists a fine tuning between ROS and antioxidant levels, which is impaired due to increased free radicals during disease conditions. AD (Alzheimer’s disease), PD (Parkinson’s disease) and ALS (amyotrophic lateral sclerosis) are classical neurodegenerative diseases caused mainly by excess free radicals. The attenuation of mitochondrial complex I activity is a major cause of ROS generation in PD patients. In PD models, ROS is mainly generated by the accumulation of aberrant mitochondria [17]. Elevated ROS levels and amyloid plaques are directly associated in the pathogenesis of AD in transgenic mice and in the postmortem brains of AD patients [18]. The ketoglutarate dehydrogenase enzyme complex controls oxidative stress and ROS production. Reduced α-ketoglutarate dehydrogenase enzyme complex activity is observed in AD brains [19]. About 20% of familial ALS cases are caused due to mutations in the *SOD1* (superoxide dismutase 1) gene [20]. The expression of mutant SOD1 is associated with distorted mitochondrial morphology as well as impaired mitochondrial function [21].

### 2.3. Intracellular Translocation of Mitochondria in Neurons

The intracellular migration of mitochondria is tightly regulated to sustain energy homeostasis and protect from oxidative stress. Miro (an outer mitochondrial membrane protein) helps in anchoring the mitochondria to microtubule-associated motor proteins and is detached to halt mitochondrial movement as an initial step in the removal of defective mitochondria. Tau, the microtubule-associated protein, constitutes insoluble filaments in the form of neurofibrillary tangles in AD and related tauopathies. Physiologically, tau controls the assembly and structural stability of microtubules. Caspase-cleaved tau modulates the translocation of mitochondria due to an increase in trafficking kinesin-binding protein 2 binding to mitochondria and a decrease in ATP production [22]. The mitochondrial SNPH protein helps in placing mitochondria along axons to match metabolic needs [23].

### 2.4. Mitochondrial Dysfunction and Neurodegeneration

The loss of normal mitochondrial function and chronic mitochondrial stress are major contributors to the pathogenesis of many neurodegenerative diseases, including AD and PD. Here, the association of defective mitochondrial function with different neurodegenerative diseases is discussed.

#### 2.4.1. Parkinson’s Disease

The dysregulation of mitochondrial transport, abnormal function and aberrant autophagy are key features observed in PD pathogenesis. The exposure of agrochemicals selectively triggers a deficit in mitochondrial transport by nitrating the microtubules in neurons harboring the synuclein, *SNCA-A53T* mutation, thus demonstrating a gene environment interaction in PD [24]. Neurotoxin-induced animal models of PD, such as 6-OHDA and MPTP, demonstrate selective targeting of complex I of the ETC, thus disrupting mitochondrial respiration of dopamine neurons in the SNpc region. Several histone modifications and mediators have been linked to age-dependent neurodegeneration. Sirtuins (NAD-dependent histone deacetylases) are involved in aging and longevity. Mechanistically, altered NAD+ metabolism in patient-derived cells contributes to Sirtuin2 activation and ensuing a reduction in the acetylated from of α-tubulin levels [25]. Aberrant mitochondrial biogenesis causes an increasing loss of dopamine neurons and motor impairments in Drosophila models of PINK1 or parkin insufficiency. Mutations in *PINK1* and *parkin* cause autosomal recessive PD, plausibly due to defective PINK1/Parkin-mediated mitophagy [26]. PD-associated *VPS35* (vacuolar protein sorting ortholog 35) mutations show mitochondrial fission and cytotoxicity in cultured neurons in vitro, in mouse substantia nigra in vivo and in human fibroblasts from PD patients with the *VPS35(D620N)* mutation [27]. Another PD-related protein, LRRK2 (leucine-rich repeat kinase 2), is reported to form a complex with Miro and accelerate its removal. In contrast, pathogenic *LRRK2G2019S* mutants abolish this task, which results in slowing down the accumulation of damaged mitochondria and, subsequently, reducing the initiation of mitophagy. The degradation of Miro protein and disabled mitochondrial movement are also diminished in sporadic cases of PD [28]. Interestingly, α-syn accumulation has been shown to induce the upregulation of Miro protein levels due to an interaction via the N-terminus of α-syn [29]. Mutations in the *DJ-1* gene were found to be causally linked to autosomal recessive PD cases. The loss of DJ-1 function leads to the perturbation of several mitochondrial functional proteins involved in the TCA cycle and electron transport chain. Mitochondrial respiration in synapses exhibits a significant alteration due to DJ-1 deficiency [30].

#### 2.4.2. Frontotemporal Dementia and Alzheimer’s Disease

FTD (frontotemporal dementia) is another neurodegenerative disease where mitochondrial quality control is markedly compromised. Overproduction of the ROS in mitochondria due to an increase in mitochondrial membrane potential in FTDP-17 causes oxidative stress and cytotoxicity [31]. In AD, mitochondria are reportedly involved in Aβ (amyloid-β) deposition. Mutation in a Norwegian family in PITRM1 (Pitrilysin metallopeptidase 1) protein results in a novel phenotype associated with Aβ accumulation [32]. Mitochondrial localizing protein SIRT3 (Sirtuin 3) protects the cell against oxidative or metabolic stress, and it has been observed that mRNA and protein levels of SIRT3 are robustly reduced in the cerebral cortex of the AD brain. Also, downregulation of SIRT3 results in p53-mediated damage to the mitochondria and neurons in AD [33]. SIRT3 exerts protection to parvalbumin and calretinin interneurons against Aβ-related degeneration in AppPs1 mice by maintaining proper mitochondrial function [34]. In another pertinent study, SIRT3-knockout mice exhibited poor remote memory, impaired long-term potentiation and decreased neuronal number in the anterior cingulate cortex, which seemed to contribute to their memory deficiencies [35]. ApoE4 has been reported to modulate mitochondrial function and metabolism during AD pathogenesis. Global proteomic analysis revealed widespread functional modifications in the mitochondria of apoE4 cells, such as diminished levels of many respiratory complex subunits and remarkable distortion to almost all subunits in complex V. In addition, apoE4 cells demonstrated notably altered levels of proteins linked to mitochondrial-associated membranes, fusion/fission and mitochondrial translocation protein [36].

#### 2.4.3. Amyotrophic Lateral Sclerosis

TDP-43 (RNA-binding protein) is a major element of protein aggregation observed in ALS and numerous other neurodegenerative diseases. Both full-length (mitochondrial matrix form) and truncated isoforms (mitochondrial intermembrane space protein) of TDP43 generate toxic aggregates, implying the occurrence of full-length TDP-43 as a main reason for the mitochondrial defect [37]. The synaptosomal fraction from the spinal cord and motor cortex of an ALS mouse model showed dysfunctional mitochondria, exhibiting hyperactive hexokinase and phosphofructokinase, citrate synthase and malate dehydrogenase [38]. C9orf72 (mitochondrial inner-membrane-associated protein) regulates cellular bioenergetics by controlling oxidative phosphorylation. In a C9orf72-ALS model, impairment of the mitochondrial function mediated axonal dysfunction [39]. Intriguingly, in C9orf72-linked patient-derived neurons, the impairment of mitochondrial complex I function was reported [40]. Optineurin exhibits close relevance to pathogenesis in neurodegeneration. In cases of ALS and FTLD, mutations in optineurin have been reported. Optineurin functions as a key autophagy receptor during mitophagy, along with the five adaptors that migrate to mitochondria [41]. A study by Fang et al. showed that the levels of mitophagy-related proteins are downregulated in AD brains and iPSC-derived cortical neuronal cultures obtained from AD patients [42].

## 3. Astrocytic Mitochondria and Their Contribution in Neurodegeneration

Astrocytes constitute about 20–40% of the brain cells, depending on the counting methodology and brain region. Due to their dynamic neuroinflammatory activity, astrocytes are generally considered as another important responder to neurodegeneration, which are evolving as leading drivers of several brain diseases. Recent evidence suggests that astrocytes may aid the development of neurodegenerative diseases due to their main role in helping towards neuronal function and metabolism [43]. Under an activated state, not only microglia but astrocytes also release soluble factors (glutamate, ROS and cytokines) that adversely influence neurons [44]. Astrocytic mitochondrial dysfunction causes neuroinflammatory triggering of astrocytes. Likewise, many key astrocytic functions, such as Ca^2+^ signaling, glutamate metabolism and antioxidant production, are governed by healthy mitochondria. Research on astrocytic mitochondrial proteins and their importance in signaling mechanisms has been correlated with neurodegeneration. Astrocytic projections form functionally isolated microdomains, thus facilitating local homeostasis by redistributing ions, removing neurotransmitters, and releasing factors to regulate blood flow and neuronal activity. Microdomains exhibit a spontaneous increase in Ca^2+^, ensuring local metabolic support to astrocytes for energetically demanding processes [45]. Defective mitochondrial biogenesis and function along with increased ROS generation are critical factors of aging. A simultaneous decrease in the dynamic levels of IGF-1 is related to aging and neurodegeneration. Astrocytic mitochondrial function is regulated by IGF-1, which is important for coordinating hippocampal-dependent spatial learning; thus, a decrease in IGF-1 receptors with age is linked with reduced learning and enhanced gliosis [46]. It has been suggested that defective mitochondrial homeostasis in astrocytes contributes to neurodegeneration. The m-AAA (mitochondrial AAA) proteases exert quality and functions vital for mitochondrial homeostasis. *AFG3L2*, encoding one of the m-AAA protease subunits, is mutated in spinocerebellar ataxia and in infantile syndromes. Astrocytic malfunctioning amplifies both neuroinflammation and glutamate excitotoxicity in patients with the AFG3L2 mutation, which contributes to neuronal death [47]. FTD3 (frontotemporal dementia type 3) is amongst the most prevalent early-onset dementias, which is clinically, pathologically and genetically heterogeneous. Markedly, FTD3 patient-derived astrocytes demonstrated the dysregulation of glutamate–glutamine homeostasis, impaired mitochondria function and glutamine hypermetabolism [48]. In another study on FTD3, it was shown that a point mutation in the CHMP2B (charged multivesicular body protein 2B) causes aberrant autophagy that, in turn, triggers disconcerted mitochondrial dynamics with poor glycolysis, enhanced ROS and stretched mitochondria. This alteration in astrocytic mitochondrial function affects the phenotype of reactive astrocytes and enhanced secretion of toxins, which are stored in the NF-κB pathway, leading to neurodegeneration [49] (Figure 2).

In ALS, reactive astrocytes facilitate neurodegeneration and microglia activation by modulating their expression and toxin release. Replacement of miR-146a in the cortical astrocytes of mSOD1 (symptomatic SOD1-G93A) with dipeptidyl vinyl sulfone or by pre-miR-146a revokes their phenotypic deviances and paracrine deleterious effects to motor neurons and microglia [50]. Astrocytes and motor neurons are linked to ALS pathogenesis. In ALS patient-specific VCP (valosin-containing protein)-mutant MNs, a significant increase in cytoplasmic TDP-43 (TAR DNA-binding protein 43) levels and ER (endoplasmic reticulum) stress was observed as primary pathogenic events followed by secondary events involving impaired mitochondria [51]. One study reported that 3-nitropropionic acid (a mitochondrial toxin) treatment induces the secretion of soluble neuroprotective factors in response to BDNF in striatal astrocytes expressing mutant huntingtin [52]. Under normal physiological conditions, astrocytes undergo internalization and clear neuronal mitochondria in a specialized manner called trans mitophagy. In AD mouse brains, a significant increase in the internalization of neuronal mitochondria in astrocytes and increased degradation of neuronal mitochondria by astrocytes might involve S100a4 in the diminished transfer of mitochondria between neurons and AD astrocytes along with a significant increase in ROS in old AD astrocytes [53]. FA (Friedreich’s ataxia) is a recessive, predominantly neurodegenerative disorder caused by mutations in the first intron of the *Fxn* (frataxin) gene. Fxn (a ubiquitous mitochondrial protein) is involved in iron–sulfur complex biogenesis, and a reduction in Fxn protein levels causes the symptoms detected in this disease. In vitro knockdown of Fxn leads to the downregulation of both mRNA and protein expression, together with a mitochondrial superoxide increase and report signals of p53-mediated cell cycle arrest and apoptosis in neuron glia interaction [54]. mtDNA depletion in astrocytes attenuates their primary cilium. The depletion of oxidative phosphorylation in astrocytes promotes FOXJ1 and RFX transcription factors known as key regulators of ciliogenesis. During stress, the primary cilium elongates and becomes remarkably distorted due to a chronic stimulation of the mitochondrial stress signal in astrocytes that initiates the anabolic metabolism and advances ciliary elongation, suggesting a novel pathogenic mechanism for mitochondria-related neurodegeneration [55].

## 4. Microglial Mitochondria and Their Significance in Neurodegeneration

Microglia are potent plastic cells in the brain, which carry out widespread specialized tasks in the CNS that demand high energy from mitochondria. These cells are extremely metabolically flexible and capable of reorganizing their mitochondrial function triggered due to inflammation. Activated microglia promote a neurotoxic, inflammatory environment in the mammalian CNS that drives the pathogenesis of neurodegenerative diseases. The accumulation of microglial mtDNA oxidative damage and an increase in intracellular ROS production activate the redox-sensitive NF-kB to enhance neuroinflammation (Figure 2). The stimulation of microglia by inflammasomes leads to elevated ROS levels and the loss of mitochondrial membrane potential and integrity [56]. Mitochondrial ROS and cathepsin B are necessary for the production of interleukin-1β in microglia, a key inflammatory cytokine. Supporting the neuroinflammatory role of microglia in neurodegeneration, one of the studies demonstrated an enhanced expression of cytokines TGFβ1 in microglia and an aggravated inflammatory response, Smad3 signaling to pathological alterations in aged mice [57]. ROS is mainly produced by NOX2 in microglia, and the activation of NOX2 is linked to mitochondrial DAMP (damage-associated molecular pattern) signaling, inflammation and amyloid plaque deposition. Furthermore, ROS produced from NOX and mitochondria might work like secondary messengers and transmit immune activation and, hence, intracellular ROS signaling may cause extreme inflammation and oxidative stress [58]. In a crosstalk study between neuronal and non-neuronal cells, mitochondrial DAMPs were shown to initiate proinflammatory immune responses from non-neuronal glial cells, including microglia and astrocytes, thereby elucidating their significant contribution to chronic neuroinflammation and accelerating neurodegeneration [59]. The NLRP3 inflammasome signaling pathway is a major contributor to the neuroinflammatory process in the CNS. The dysfunction of microglial mitochondria can strengthen NLRP3 inflammasome signaling to enhance dopaminergic neurodegeneration [60]. The release of fragmented and dysfunctional microglial mitochondria has been shown to activate naive astrocytes to the A1 state, which triggers neuronal death, hence causing neurodegeneration [61]. In reactive astrocytes, mitochondrial DAMPs activate the NLRP3 inflammasome, liberating IL-1β and IL-18 via gasdermin D pore, a key pathway to boost inflammation (Figure 2). Several preclinical models, including the DBA/2J mouse (model of secondary glaucoma) and patient data, link mitochondrial loss and inflammation in glaucomatous neurodegeneration [62]. Inducible microglial mitochondrial fission has been reported to promote NLRP3-dependent neuroinflammation in hereditary and idiopathic PD [63]. In postnatal organotypic hippocampal slice cultures, microglia undergo transformation into a discrete phenotype through single toll-like receptor 4 stimulation with LPS (lipopolysaccharide), which is altered in the process of inhibition of the respiratory chain [64]. Mitochondrial UCP2 (uncoupling protein-2) controls mitochondrial functions and is involved in a variety of physiological and pathological processes. Upon LPS exposure of UCP2-silenced microglia, there is an activation of enhanced inflammatory response, depicted by a remarkable expression of *M1* genes, while sparing *IL-4* (interleukin-4) in inducing *M2* genes [65].

## 5. Significance of Oligodendrocytic Mitochondria in Axonal Dysfunction and Neurodegeneration

Oligodendrocytes express monocarboxylate transporter 1 (MCT1) at the abaxonal and adaxonal myelinic channels. Oligodendrocytes accumulate intracellular lactate that can exchange with MCT1 in the periaxonal space, where neurons can transfer it through MCT2 and metabolize lactate to match their metabolic demands [66]. Adult oligodendrocytes sustain ATP homeostasis with glycolysis, and stored lactate in oligodendrocytes can either be consumed to help myelin lipid synthesis or provide lactate to neurons [67]. The distorted oligodendrocyte-axon unit in traumatic injuries, AD and MS results in axonal dysfunction, ultimately leading to neurodegeneration. MS is a chronic demyelinating disease of the CNS and a leading cause of neurological disability globally. In general, MS is a heterogeneous autoimmune disease characterized by inflammation, demyelination and axonal degeneration, affecting both the white and gray matter of the CNS. Mitochondria have been increasingly linked to the pathogenesis of MS. Recent studies indicate a role of mitochondrial dysfunction in the neurodegenerative aspects of MS [68]. Research studies on mitochondrial dysfunction in MS mainly focused on neurons, with no reports exploring the dysregulation of mitochondrial bioenergetics and/or the genetic aspects of oligodendrocytes that might be linked with the etiopathogenesis of MS and relevant demyelinating syndromes. Large-scale epidemiological and genomic studies have unraveled several genetic and environmental risk factors of MS. Mitochondria-targeted antioxidants like SkQ1 could have promising therapeutic potential for MS and related neurological disorders [69]. Oxidative stress in oligodendrocyte’s mitochondria indicates an early sign of MS-related inflammation and reveals that evolving redox and morphological changes in mitochondria are associated with oligodendrocyte dysfunction in neuroinflammation [70]. Another classical neurodegenerative disease in the context of oligodendrocytes, i.e., MSA, is neuropathologically characterized by α-syn aggregates in oligodendroglia and clinically characterized by parkinsonism, ataxia and autonomic dysfunction. Since abnormal mitochondrial health due to oxidative phosphorylation is reported in MSA, mitochondrial haplogroup background may present a risk for MSA [71]. HSP (hereditary spastic paraplegia) is characterized by the degeneration of CNS axons. PLP (Myelin proteolipid protein) and axon-enriched proteins are mutated proteins that are involved in mitochondrial function and SER (smooth endoplasmic reticulum) structure. In HSP pathogenesis, juxtaparanodal mitochondrial deterioration, decreased mitochondria–SER associations and reduced ATP production induce axonal ovoid formation and axonal degeneration [72]. Oligodendrocyte-specific mtDNA double-strand breaks in a PLP/mtPstI mouse model trigger oligodendrocyte death and demyelination related to axonal injury and glial activation [73]. VDAC1 is a mitochondrial porin implicated in the cellular metabolism and apoptotic pathway in several neuropathological events. In spinal cord injuries, primary cell death subsequently leads to the release of pro-inflammatory molecules to trigger apoptosis followed by inflammation and demyelination, causing the loss of motor functions [74]. In oligodendrocytes, the overactivation of AMPARs (AMPA-type ionotropic glutamate receptors) induces intracellular Ca^2+^ overload and excitotoxic death. It has been shown that AMPAR activation triggers Drp1-mediated mitochondrial fission in oligodendrocytes [75]. Genetically induced mitochondrial or myelin dysfunction experimentation validates that modification in axonal mitochondrial size depends on the thickness of the myelin sheath [76]. Under an unfavorable bioenergetic scenario, iron-deficient oligodendrocytes fail to undergo myelination, thus indicating that management of the cell metabolic rate might effect the overall growth of cells [77]. The loss of MnSOD (Mn-Superoxide dismutase) in the spinal cord fosters a phenotypic alteration of inflammation, demyelination and progressive paralysis that represents phenotypes correlated with progressive MS [78].

## 6. Conclusions

A healthy mitochondrial population governs neuronal health, while accumulated dysfunctional mitochondria are a classical feature of neurodegeneration. Neurodegenerative diseases exhibit compromised mitochondrial bioenergetics, elevated oxidative stress and alterations in calcium homeostasis. The cumulative influence of dysfunctional mitochondria at different sites and cell types is correlated to divergent neuropathologies. A better understanding of the causal association between distinct modifications in mitochondria during disease progression could lead to the development of novel therapeutic strategies for neurodegenerative diseases.

## Figures and Tables

**Figure 1 ijms-24-14421-f001:**
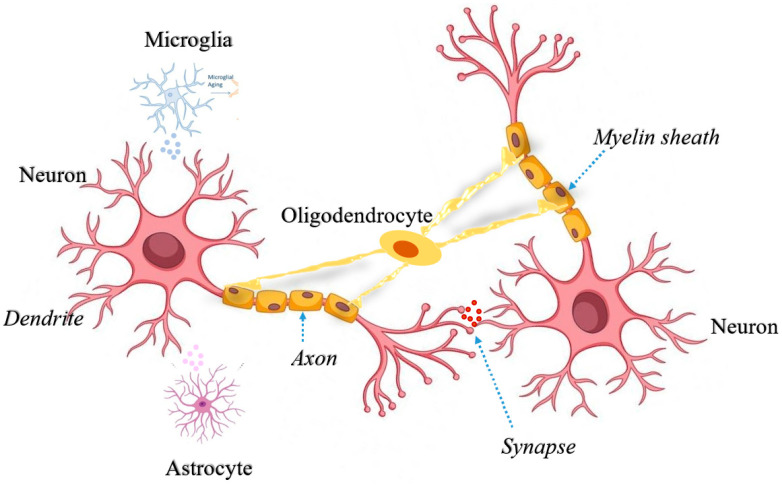
Interaction among diverse cell types in the brain. Secretory factors released from activated microglia and reactive astrocytes trigger neuroinflammatory signals in the neurons. Oligodendrocytes support neuronal axons for smooth neurotransmission from dendritic spines. All different types of cells are represented with different colors and specialized networks among these cells depict the fine tuning to support neuronal health and transmission.

**Figure 2 ijms-24-14421-f002:**
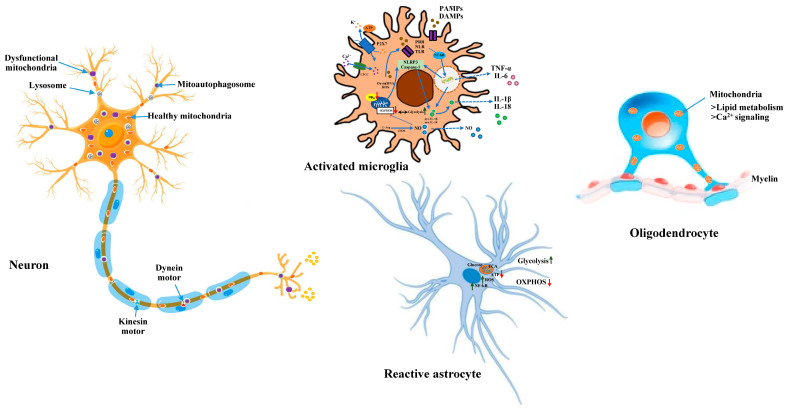
Mitochondrial dynamics in neurons, microglia, astrocytes and oligodendrocytes. Distinct mitochondrial states, both structural and functional, in different cell types are represented along with key molecular events of oxidative stress and bioenergetics. Abbreviations used are: PAMPs, pathogen-associated molecular patterns; ATP, adenosine triphosphate; DAMPs, damage-associate molecular patterns; Δψ_m_, mitochondrial membrane potential; IL, interleukin; iNOS, inducible nitric oxide synthase; L-Arg, L-arginine; LTCC, L-type Ca^2+^ channel; NF-kB, nuclear factor kappa B; NLR, NOD like receptors; NLRP3, NLR family, pyrin domain containing 3; NO, nitric oxide; OXPHOS, oxidative phosphorylation; PRR, pattern-recognition receptors; P2X7, purinergic receptor P2X7; ROS, reactive oxygen species; TCA, tricarboxylic acid; TLR, toll-like receptors; TNF-α, tumour necrosis factors-α.

## Data Availability

All data is available on NCBI pubmed.

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
