# Peer review of "Cell-Type-Specific Mitochondrial Quality Control in the Brain: A Plausible Mechanism of Neurodegeneration"

_ijms, 2023, doi:10.3390/ijms241914421_

Round 1
Reviewer 1 Report
In the present manuscript entitled “Cell-type specific mitochondrial quality control in the brain: a precise mechanism of neurodegeneration”, Ragupathy et al. discussed that mitochondria are extremely important in maintaining the physiological functions of nervous system, and their dysfunction can lead to neurodegeneration diseases. In particular, they focused on the mitochondrial functions and signaling pathways in the glial cells, astrocytes, microglia, and oligodendrocytes. It is well summarized what kind of diseases are caused by mitochondria dysfunction and by what mechanism they are regulated. Also, the authors cite appropriate references. Following are few minor comments for this study.
Minor issues:
1) Figure 2 is very well organized, but the signaling molecules and pathways in the cell are too small to see, so please improve, if possible.
2) Please check the references, as some years are in bold and some are in fine print.
3) Please specify what SNPH and HSP stand for.
4) NLRP3 (NOD-like receptor family pyrin domain-containing 3) on page 6 have different fonts.
Author Response
We thank the reviewer for the nice comments to our review article. This would add an important insight on the mitochondrial health and mechanisms of neurodegeneration in different brain cells. Below is the pointwise reply to the minor issues raised by the reviewer: Minor issues: 1) Figure 2 is very well organized, but the signaling molecules and pathways in the cell are too small to see, so please improve, if possible. Reply: In Fig. 2, we tried to use the best possible font size and shape for each cell types, as we included only key signaling molecules and pathways. Considering the limitation due to vast signaling cascade in each cell type, we were unable to further enlarge the font size. 2) Please check the references, as some years are in bold and some are in fine print. Reply: We have checked the references thoroughly and fixed the font style as per reviewer’s suggestion. 3) Please specify what SNPH and HSP stand for. Reply: We have mentioned the abbreviation for SNPH i.e. syntaphilin (line#42) and HSP i.e. hereditary spastic paraplegia (line#433). 4) NLRP3 (NOD-like receptor family pyrin domain-containing 3) on page 6 have different fonts. Reply: We have fixed the fonts as suggested by the reviewer (line#379).Reviewer 2 Report
The topic is definitely interesting and the review covers a plethora of data concerning the relationship of mitochondria and the various cell types in neurodegeneration.
However, it is not well-organised and difficult to follow, especially the introduction. I would suggest a shortened introduction, then the role of mitochondria in the different cell types in health and neurodegeneration could be discussed in details in sections 2-5. Maybe it would be worth describing first the neurons, then the glial cells. It is difficult to follow the alterations in the distinct neurodegenerative diseases in section 4: there is tau (AD), then PD, FTD, then beta-amyloid (AD) again.
line 472-473: “The loss in mitochondrial energy production, oxidative stress, and changes in calcium handling are associated with neurodegenerative diseases”. Indeed, neurodegenerative diseases are often associated with mitochondrial dysfunction. But mitochondrial dysfunction can be found in other diseases as well: kidney disease (PMID: 36839892), or Type 2 Diabetes Mellitus (PMID: 36866104) and many more diseases (PMID: 36830595).
The title is a little exaggerates: “a precise mechanism of neurodegeneration”.
What kind of mouse is the DBA/2J? (line 248)
Caspase-cleaved tau is mentioned (line 301), but the role of tau in Alzheimer’s disease is not discussed.
Drosophila models of PINK1 is described (line 317), but the connection of PINK1 to PD is mentioned only in line 342.
line 403: HSP7, line 448: only HSP.
It is mentioned shortly that oligodendrocytes support neuronal energy metabolism (line 425), but I feel that more details about the lactate shuttle may be beneficial (PMID: 28862639).
Author Response
We thank the reviewer for positive comments to our review article and valuable suggestions to re-organize and elaborate some important topics relevant to the scope of this review article. As per suggestion, we have exhaustively re-organized the introduction section and shortened it by including the most relevant literature information. Also, in accordance to the reviewer’s suggestion, we have discussed neurons first in section 2, followed by astrocytes, microglia, then oligodendrocytes in section 3, 4 and 5, respectively. Additionally, we have subdivided section 2 into sub-sections, so as to make the content simpler and easy to follow.
It is difficult to follow the alterations in the distinct neurodegenerative diseases in section 4: there is tau (AD), then PD, FTD, then beta-amyloid (AD) again.
Reply: We thank the reviewer for pointing out on this, now we have re-organized section 2 (original version section 4) in our revised version of the manuscript.
line 472-473: “The loss in mitochondrial energy production, oxidative stress, and changes in calcium handling are associated with neurodegenerative diseases”. Indeed, neurodegenerative diseases are often associated with mitochondrial dysfunction. But mitochondrial dysfunction can be found in other diseases as well: kidney disease (PMID: 36839892), or Type 2 Diabetes Mellitus (PMID: 36866104) and many more diseases (PMID: 36830595).
Reply: We completely agree with the reviewer’s comment on association of neurodegenerative diseases with mitochondrial dysfunction and in case of other diseases. We have edited the line 472-473 accordingly, which is now changed to line 458-459 in the revised version.
The title is a little exaggerates: “a precise mechanism of neurodegeneration”.
Reply: We agree with the reviewer’s comment. We have modified the title as per suggestion.
What kind of mouse is the DBA/2J? (line 248)
Reply: DBA/2J mouse strain is a well-characterized model of secondary glaucoma arising due to iris pigment dispersion and atrophy. We have added this information in line#388. Please note that the line# is changed from 248 to 388 in the revised version.
Caspase-cleaved tau is mentioned (line 301), but the role of tau in Alzheimer’s disease is not discussed.
Reply: We have incorporated this key information about the role of tau in AD in the revised version in line#167-169.
Drosophila models of PINK1 is described (line 317), but the connection of PINK1 to PD is mentioned only in line 342.
Reply: We have re-organized the description about PINK1 and its connection to PD in the revised version in line#197-199. Please note that the line 342 is changed to line#200 in the revised version.
line 403: HSP7, line 448: only HSP.
Reply: In the revised version of our manuscript, now we only include HSP (hereditary spastic paraplegia) i.e. line#433-439.
It is mentioned shortly that oligodendrocytes support neuronal energy metabolism (line 425), but I feel that more details about the lactate shuttle may be beneficial (PMID: 28862639).
Reply: As suggested by the reviewer, we have elaborated (in section 5) the significance of lactate shuttle mediating oligodendrocyte support to neuronal energy demand. We have incorporated this information in the revised version in line#405-411.
Reviewer 3 Report
In this manuscript authors review the relevance of mitochondrial quality control in neuronal survival. They also point-out the importance of maintaining healthy mitochondria in astrocytes, oligodendrocytes and microglia since their dysfunction plays an important role in the development of neurodegenerative diseases. Authors also include an interesting item that summarizes an important number of proteins that regulate mitophagy, mitochondrial fission/fusion dynamics, intracellular migration, or mitochondrial biogenesis, which are mechanisms involved in the proper functioning of mitochondria. Because the dysfunction of these mechanisms promotes mitochondrial impairment, authors propose that. mitochondria could be an interesting target to develop new strategies for the treatment of neurodegenerative diseases.
The manuscript is well written and organized, and all the information included is clear and concise. The bibliography is extensive and actualized. To this reviewer, there is no minor or mayor concerns regarding this manuscript.
Author Response
We thank the reviewer for the nice and encouraging comments on our review article with no additional edits/queries suggested.
Reviewer 4 Report
The present review by Ragupathy et al. is an interesting article that highlights the importance of mitochondrial quality control in brain along its critical role in neurodegeneration. The role of cell specific mitochondrial quality control makes it more interesting; however, I have a few queries:
Please rectify the typological errors throughout the script.
Line No 58: Please add the reference.
Line No: 86: Please add the reference.
Line No. 89: Please fix the spacing.
Line No. 99: Please add the reference to support the information provided.
Line No. 206: Please fix the spacing.
Line No. 239: Please fix the font size.
Line No. 319: Please check the font size
Line 331: Please fix the spacing.
Section 4 is too lengthy. Authors should split it into several paragraphs to maintain the flow of reading.
Authors have provided a detailed description of mitochondrial dysfunction in every cell type but have not provided any reason governing the defective mitophagy. The authors should focus on that part to describe the mechanistic part.
One major reason is the mutation and/or degradation of different mitophagy receptors like optineurin, BNIP3, BNIP3L, FUNDC1 etc. The authors should add this information in context to different cell types that the authors have described.
In addition to the loss of ATP production ability, Another major pathology associated with the mitochondrial dysfunction is the induction of oxidative stress by mitochondria generated superoxide radicals. The authors have gently touched this information but have not provided any detailed information in context to different cell types. The authors should also focus a little bit more on this part.
Different neuropathogenic bacteria and virus also have the ability to disrupt the clearance of defective mitochondria. The infection of brain cells with these pathogens is also projected to be an important tool that accelerates neurodegenerative diseases. What are the authors views on this? The authors should also list it as one of the factors that causes mitochondrial dysfunction in brain.
Author Response
We thank the reviewer for the positive comments and suggesting minor typographical errors in our manuscript. Following are the pointwise reply to the edited errors and queries:
Line No 58: Please add the reference.
Reply: This sentence has been deleted while revising the introduction section as suggested by another reviewer.
Line No: 86: Please add the reference.
Reply: As the content of line no. 86 is in relevance to line no. 87-89, reference no. [7] Jeon et al. 2020 is included in line no. 89.
Line No. 89: Please fix the spacing.
Reply: This sentence has been deleted in our revised version of the manuscript.
Line No. 99: Please add the reference to support the information provided.
Reply: We have added the relevant reference as per reviewer’s suggestion in revised version in line#44. Please note that the line#99 is changed to line#44 in our revised version of the manuscript.
Line No. 206: Please fix the spacing.
Reply: Spacing has been fixed as per reviewer’s suggestion. Please note that Line#206 is changed to line#346 in our revised version of the manuscript.
Line No. 239: Please fix the font size.
Reply: Font size has been fixed as per reviewer’s suggestion. Please note that Line#239 is changed to line#379 in our revised version of the manuscript.
Line No. 319: Please check the font size
Reply: Font size has been corrected as per reviewer’s suggestion. Please note that Line#319 is changed to line#138 in our revised version of the manuscript.
Line 331: Please fix the spacing.
Reply: Spacing has been fixed as per reviewer’s suggestion. Please note that Line#331 is changed to line#184 in our revised version of the manuscript.
Section 4 is too lengthy. Authors should split it into several paragraphs to maintain the flow of reading.
Reply: We thank reviewer for suggesting re-organization of section 4. We have sub-divided section 4 into several paragraphs, now refer to section 2 (sub-sections 2.1, 2.2, 2.3, 2.4, 2.4.1, 2.4.2, 2.4.3). This has substantially helped our manuscript to maintain the flow of reading.
Authors have provided a detailed description of mitochondrial dysfunction in every cell type but have not provided any reason governing the defective mitophagy. The authors should focus on that part to describe the mechanistic part.
Reply: We thank the reviewer for nice comment and bringing this point of mechanistic part for the regulation of defective mitophagy in every cell type. The focus of this review article is to cover the literature on the mitochondrial quality control in different brain cells and the overall contribution of non-neuronal cells to maintain the neuronal health. Therefore, a detailed discussion of the cell type specific mitophagy events is out of the scope and thus not included in this review article. However, we are looking forward to preparing another exhaustive review mainly focusing on mechanistic insight of mitophagy in different brain cells.
One major reason is the mutation and/or degradation of different mitophagy receptors like optineurin, BNIP3, BNIP3L, FUNDC1 etc. The authors should add this information in context to different cell types that the authors have described.
Reply: We thank the reviewer for this valuable suggestion that would further provide mechanistic insight. We have included this key information in the revised version of our manuscript in line#257-265 by including relevant references. In context to different cell types, we are excited to prepare another review article focusing mainly on this interesting mechanistic aspect in different cell types in the brain.
In addition to the loss of ATP production ability, Another major pathology associated with the mitochondrial dysfunction is the induction of oxidative stress by mitochondria generated superoxide radicals. The authors have gently touched this information but have not provided any detailed information in context to different cell types. The authors should also focus a little bit more on this part.
Reply: We thank the reviewer for this valuable suggestion that would further provide mechanistic insight. We have included this key information in the revised version of our manuscript in line#145-159 by including relevant references. In context to different cell types, we are excited to prepare another review article focusing on mitochondrial oxidative stress in the brain.
Different neuropathogenic bacteria and virus also have the ability to disrupt the clearance of defective mitochondria. The infection of brain cells with these pathogens is also projected to be an important tool that accelerates neurodegenerative diseases. What are the authors views on this? The authors should also list it as one of the factors that causes mitochondrial dysfunction in brain.
Reply: We thank the reviewer for bringing this very important and emerging hot topic of neuropathogenic agents including bacteria and virus that definitely has potential impact on disrupting mitochondrial quality control and hence, play key role in accelerating pathogenesis of neurodegenerative diseases. As our focus in the present review article is to discuss the classical/fundamental aspect of mitochondrial quality control in different brain cells, we would be thrilled to prepare another review article focusing on the advancement on microbial infection and neuroinflammation in mitochondrial quality check in brain and neurological disorders.
Round 2
Reviewer 2 Report
The authors have adequately answered my questions and comments.
Reviewer 4 Report
The authors have addressed all the comments and the script has been removed. I recommend the acceptance.